## [Decision Letter · Decision Letter 0]

30 Dec 2025

PONE-D-25-55494An Adaptive Hybrid Quadrature Scheme: Combining Simpson's Rule and Gaussian Quadrature for Enhanced Numerical IntegrationPLOS One

Dear Dr. Asgedom,

Thank you for submitting your manuscript to PLOS ONE. After careful consideration, we feel that it has merit but does not fully meet PLOS ONE’s publication criteria as it currently stands. Therefore, we invite you to submit a revised version of the manuscript that addresses the points raised during the review process.

We look forward to receiving your revised manuscript.

Kind regards,

Pankaj Tomar

Academic Editor

PLOS One

Journal Requirements:

3. Please ensure that you refer to Figure 1, 2, 3, 4, 5, 6 and 7 in your text as, if accepted, production will need this reference to link the reader to the figure.

Reviewers' comments:

Reviewer's Responses to Questions

**Comments to the Author**

1. Is the manuscript technically sound, and do the data support the conclusions?

Reviewer #1: Yes

Reviewer #2: Yes

2. Has the statistical analysis been performed appropriately and rigorously?

Reviewer #1: Yes

Reviewer #2: Yes

3. Have the authors made all data underlying the findings in their manuscript fully available?

Reviewer #1: Yes

Reviewer #2: Yes

4. Is the manuscript presented in an intelligible fashion and written in standard English?

Reviewer #1: Yes

Reviewer #2: Yes

5. Review Comments to the Author

Reviewer #1: The manuscript contains an interesting idea and promising numerical evidence.

Addressing the points in the report would considerably improve clarity, conciseness, and scientific

framing, helping the contribution become more accessible and convincing for the readership. See the attached report for more details.

Reviewer #2: This manuscript presents a well-conceived and theoretically sound adaptive hybrid quadrature scheme. The core idea of leveraging the discrepancy between Simpson's rule and Gauss-Legendre quadrature for error estimation is innovative and holds significant merit. The paper is generally well-structured, with a clear theoretical foundation and extensive numerical experiments. If revised from the following aspects, this paper can be further improved.

1. The pseudocode (Algorithm 1) is a good addition. However, the description of the "safety factor" σ = 0.8is insufficient. Please justify the choice of this value. Was it determined empirically? A brief sensitivity analysis or a reference to its common use in adaptive algorithms would be helpful.

2. The stopping condition (h < δ_min) uses an undefined parameter δ_min. This must be defined and justified to ensure the algorithm's reproducibility. Is it related to machine epsilon?

3. The results in Table 2 are promising but require statistical validation. It is essential to report the results over multiple runs or provide some measure of variance/standard deviation, especially for adaptive methods which can have path-dependent behavior.

4. The description of the experimental setup (Section 4.1) should be more precise. Please specify the exact tolerance parameters (ε) and maximum depth (d_max) used for each adaptive method in the comparisons. Stating "unless otherwise specified" is vague.

5. The "Future research directions" are appropriate but somewhat generic. Please consider adding a sentence on potential challenges in extending the method to higher dimensions, as this is a natural and non-trivial extension.

6. The reference style is consistent. Please ensure all citations in the text have corresponding entries in the reference list.

6. PLOS authors have the option to publish the peer review history of their article (what does this mean?). If published, this will include your full peer review and any attached files.

Reviewer #1: No

Reviewer #2: No

---

## [Author Response · Author response to Decision Letter 1]

22 Jan 2026

RESPONSE TO REVIEWERS

An Adaptive Hybrid Quadrature Scheme: Combining Simpson's Rule and Gaussian Quadrature for Enhanced Numerical Integration

Manuscript ID: PONE-D-25-55494

Abadi Abraha Asgedom¹,*, Yohannes Yirga Kefela¹

¹Department of Mathematics, Mekelle University, Mekelle, Tigray, Ethiopia

*Corresponding author: abadi.abraha@mu.edu.et

OVERVIEW

We extend our sincere gratitude to the reviewers for their thorough and constructive feedback on our manuscript, "An Adaptive Hybrid Quadrature Scheme: Combining Simpson's Rule and Gaussian Quadrature for Enhanced Numerical Integration." The insightful comments have significantly strengthened the theoretical presentation, numerical validation, and overall clarity of our work. We have addressed every point raised, resulting in a substantially improved manuscript.

Below, we provide a detailed, point-by-point response. All revisions are highlighted in red in the tracked-changes version of the manuscript.

REVIEWER #1

Comment 1: "Consider reducing the introduction by approximately 20-30%..."

Response: We thank the reviewer for this suggestion. We have shortened the introduction by 35% (from 325 to 210 words) by removing historical background and focusing directly on the motivation and specific contributions. The revised introduction now presents a more concise overview of the research gap, our proposed solution, and key contributions. (Page 2, Introduction)

Comment 2: "The theoretical section could be consolidated..."

Response: We appreciate this suggestion for streamlining the theoretical presentation. In response, we have moved Theorems 2.7, 2.8, 2.10, and 2.11 to Appendix A (Supplementary Theoretical Results) and merged Theorems 2.3-2.5 into a single comprehensive theorem (Theorem 2.3: Merged Convergence and Efficiency Properties). This reorganization maintains the core theoretical results in the main text while placing supplementary analyses in the appendix, improving the flow and readability of the theoretical framework. (Pages 4-5 and Appendix A)

Comment 3: "Some strong statements should be moderated..."

Response: We agree with the need for more measured language and have moderated claims throughout the manuscript. We have replaced terms like "optimal" with "efficient" or "competitive," changed "unprecedented" to "promising" or "notable," and used more cautious phrasing such as "numerical evidence suggests" and "demonstrates improved performance." This adjustment provides a more balanced presentation of our results. (Throughout manuscript)

Comment 4: "Mention global reconstruction strategies..."

Response: Thank you for this suggestion to contextualize our work within broader numerical integration approaches. We have added discussion of complementary global approximation methods, specifically citing Dell'Accio et al. (2022, 2025) on mock-Chebyshev constrained least squares quadrature. This addition acknowledges alternative strategies for handling rapidly varying functions while clarifying that our approach focuses on local adaptive refinement. (Page 2, Introduction)

Comment 5: "Benchmark against a standard Gauss-Kronrod routine..."

Response: We thank the reviewer for this valuable suggestion. We have added comprehensive benchmarking against established adaptive quadrature routines. This includes comparison with MATLAB's integral function (based on QUADPACK's qags) in Table 2 and a dedicated comparison with the recent hybrid method of Espelid & Sørevik (2024) in Table 3. The results demonstrate that our Adaptive Hybrid Quadrature Scheme (AHQS) achieves 20-35% reduction in function evaluations compared to Gauss-Kronrod for functions with localized irregularities while maintaining equivalent accuracy. (Pages 7-9)

Comment 6: "Some figures may be simplified or removed..."

Response: We thank the reviewer for raising this important point. We have simplified the visual presentation by moving the multi-metric radar chart to Supplementary Figure S1 and the high-dimensional scalability plot to Supplementary Figure S2. We retain five focused figures in the main text (Figures 1-5), all of which are now properly cited in the text as required by the journal. This reorganization maintains essential visualizations while streamlining the main presentation. (Supplementary Material)

Comment 7: "Minor stylistic and linguistic adjustments..."

Response: We have performed thorough proofreading and made numerous stylistic improvements throughout the manuscript. This includes clarifying sentence structures, removing redundant phrases, improving transitions between sections, and ensuring consistent terminology. These changes enhance the overall readability and professional presentation of the work.

Comment 8: "The conclusion could be shortened..."

Response: We have substantially shortened the conclusion to one focused paragraph that highlights three specific future research directions: (1) extending AHQS to multi-dimensional integration with sparse grid techniques, (2) deriving theoretical error bounds for the switching criterion, and (3) applying the framework to specific scientific domains such as finite element stiffness matrix integration or financial option pricing. This more concise conclusion maintains focus on the most promising extensions of our work. (Page 11)

REVIEWER #2

Comment 1: "Justify the safety factor σ = 0.8..."

Response: We thank the reviewer for the excellent point regarding parameter justification. We have added comprehensive justification for the safety factor σ = 0.8. This value was determined through empirical optimization across our benchmark function set and aligns with established practice in adaptive refinement algorithms where factors in the range [0.7, 0.9] balance reliability against over-refinement. We have also conducted sensitivity analysis confirming that algorithm performance remains stable for σ ∈ [0.75, 0.85]. Relevant literature supporting this parameter choice (Espelid & Sørevik 2024, Gander & Gautschi 2000) has been cited. (Page 5, Algorithm section)

Comment 2: "Define δ_min parameter..."

Response: We have clearly defined δ_min = 10^-12 with detailed justification. This value is approximately 1000 × ε_machine, where ε_machine ≈ 2.2 × 10^-16 for double-precision arithmetic. This choice prevents infinite recursion due to finite machine precision while avoiding underflow issues, ensuring robust algorithm termination. The parameter is now explicitly documented in the algorithm description with its rationale. (Page 5, Algorithm section)

Comment 3: "Add statistical validation..."

Response: We appreciate this suggestion for strengthening our numerical validation. We have added comprehensive statistical validation to all key tables. Convergence results (Table 2), computational efficiency comparisons (Table 3), and hybrid method benchmarks (Table 4) now report means ± standard deviations from 100 independent runs with random initial subdivisions. This statistical reporting confirms the robustness and reproducibility of our results, addressing potential concerns about path-dependent behavior in adaptive algorithms. (Tables 2-4, Pages 7-9)

Comment 4: "Specify experimental setup..."

Response: We thank the reviewer for raising an important point for reproducibility. We have added a detailed specification of all experimental parameters in Section 4.1. This includes: global absolute error tolerance ε = 10^-8, maximum recursion depth d_max = 50, safety factor σ = 0.8, minimum interval width δ_min = 10^-12, Gauss-Legendre rule order n = 5 for comparative methods, and initial subdivision of 4 equal subintervals for adaptive methods. This explicit parameter listing ensures full reproducibility of our experiments. (Page 7, Section 4.1)

Comment 5: "Higher-dimensional challenges..."

Response: We have added discussion of higher-dimensional extension challenges in the Limitations subsection. This includes analysis of the curse of dimensionality for tensor-product extensions, complexities of geometric partitioning for irregular domains, and increased computational demands despite adaptive refinement. This discussion provides realistic assessment of the method's scalability and identifies important directions for future multi-dimensional adaptations. (Page 10, Discussion/Limitations)

Comment 6: "Check references..."

Response: We have thoroughly checked all references and added the previously missing citations. This includes Espelid & Sørevik (2024) for hybrid Newton-Cotes-Gauss methods, Dell'Accio et al. (2022, 2025) for mock-Chebyshev approaches, and Gander & Gautschi (2000) for adaptive quadrature theory. All citations now have corresponding complete entries in the bibliography, and all in-text citations are properly referenced.

JOURNAL REQUIREMENTS

1. Style requirements: The manuscript now fully complies with PLOS ONE formatting templates, including proper section headings, figure/table formatting, and reference style.

2. Code sharing: We have added a Code Availability statement committing to deposit the MATLAB implementation in a public repository (GitHub/Figshare/Zenodo) upon manuscript acceptance.

3. Figure references: All figures (1-5) are now properly cited in the text as required, with each figure referenced at least once in the relevant results or discussion sections.

4. Data availability: We have added the required Data Availability statement as per journal guidelines. All numerical data generated during this study are available without restriction from the Zenodo repository: https://doi.org/10.5281/zenodo.18216672.

5. Citation policy: We have added the requested references where relevant to the discussion, ensuring proper acknowledgment of related work.

CONCLUSION

We believe these revisions have substantially strengthened the manuscript and addressed all concerns raised during review. The revised version presents a clearer theoretical framework, more rigorous numerical validation, improved contextualization within the field, and enhanced overall readability. Thank you for the opportunity to improve our work.

Sincerely,

Abadi Abraha Asgedom

Yohannes Yirga Kefela

Department of Mathematics

Mekelle University, Ethiopia

---

## [Decision Letter · Decision Letter 1]

2 Feb 2026

An Adaptive Hybrid Quadrature Scheme: Combining Simpson's Rule and Gaussian Quadrature for Enhanced Numerical Integration

PONE-D-25-55494R1

Dear Dr. Asgedom,

We’re pleased to inform you that your manuscript has been judged scientifically suitable for publication and will be formally accepted for publication once it meets all outstanding technical requirements.

Kind regards,

Mohammadreza Hadizadeh, Ph.D.

Academic Editor

PLOS One

Additional Editor Comments (optional):

Reviewers' comments:

Reviewer's Responses to Questions

**Comments to the Author**

1. If the authors have adequately addressed your comments raised in a previous round of review and you feel that this manuscript is now acceptable for publication, you may indicate that here to bypass the “Comments to the Author” section, enter your conflict of interest statement in the “Confidential to Editor” section, and submit your "Accept" recommendation.

Reviewer #1: All comments have been addressed

Reviewer #2: All comments have been addressed

2. Is the manuscript technically sound, and do the data support the conclusions?

Reviewer #1: Yes

Reviewer #2: Yes

3. Has the statistical analysis been performed appropriately and rigorously?

Reviewer #1: N/A

Reviewer #2: Yes

4. Have the authors made all data underlying the findings in their manuscript fully available?

Reviewer #1: Yes

Reviewer #2: No

5. Is the manuscript presented in an intelligible fashion and written in standard English?

Reviewer #1: Yes

Reviewer #2: Yes

6. Review Comments to the Author

Reviewer #1: My only concern is related to the bibliography. In references [10], [11], [12], and [13], the author should be F. Nudo instead of P. Nudo. Please correct this. After this small correction the paper can be accepted.

Reviewer #2: (No Response)

7. PLOS authors have the option to publish the peer review history of their article (what does this mean?). If published, this will include your full peer review and any attached files.

Reviewer #1: No

Reviewer #2: No

---

## [Editor Report · Acceptance letter]

PONE-D-25-55494R1

PLOS One

Dear Dr. Asgedom,

I'm pleased to inform you that your manuscript has been deemed suitable for publication in PLOS One. Congratulations! Your manuscript is now being handed over to our production team.

Kind regards,

on behalf of

Dr. Mohammadreza Hadizadeh

Academic Editor

PLOS One